Diagnostic value of Ep300 and FOXO4 proteins in acute ischemic stroke: preliminary findings from a case-control study

Bülbül Emre kkartal008@hotmail.com 1
Toker İbrahim 2
Dana Halime 3
Doğan Mehmet 4
Mercan İbrahim 1
Çoşkun Işık Burcu 1
Şener Elif Funda 3
1 Emergency Medicine, Erciyes University , Kayseri , Turkey
2 Emergency Medicine, Kayseri City Hospital , Kayseri , Turkey
3 Department of Medical Biology, Erciyes University Genome and Stem Cell Center (GENKOK), Erciyes University , Kayseri , Turkey
4 Public Health, Vocational Health College, Erciyes University , Kayseri , Turkey
Uversky Vladimir
Electronic publication date: 2025 Aug 29
Publication date: 2025
Volume: 13
Electronic Location ID: e19970
Received 2025 Mar 27; Accepted 2025 Jul 31
Copyright: ©2025 Bülbül et al.
Copyright year: 2025
Copyright holder: Bülbül et al.
License: This is an open access article distributed under the terms of the Creative Commons Attribution License, which permits unrestricted use, distribution, reproduction and adaptation in any medium and for any purpose provided that it is properly attributed. For attribution, the original author(s), title, publication source (PeerJ) and either DOI or URL of the article must be cited.
License URL: https://creativecommons.org/licenses/by/4.0/

Keywords: Ischemic stroke, Foxo4 protein, Ep300 protein, Human

Funding: The authors received no funding for this work.

==============================
Background

Stroke is a leading cause of death and disability worldwide, and there is still a lack of specific and sensitive biomarkers for its diagnosis.

Objective

This study aimed to investigate the diagnostic value of FOXO4 and Ep300 proteins in acute ischemic stroke patients who visited the emergency department.

Methods

Patients were consecutively included in the study. The amount of Ep300 and FOXO4 proteins was determined using an enzyme-linked immunosorbent assay (ELISA). Receiver operator characteristic (ROC) analyses of FOXO4 and Ep300 proteins were performed.

Results

The study was conducted on a total of 39 acute ischemic stroke patients, 17 females and 22 males, with a mean age of 66.9 ± 11 years. Seventeen females and 23 male control were also included. The discriminative ability of Ep300 protein was not statistically significant (p value = 0.380). FOXO4 protein had moderate discriminative ability (AUC value = 0.705 and p value = 0.002). When the cut-off value for FOXO4 protein was accepted as > 1.15, the sensitivity was 74.29%, the specificity was 64.52%, the positive predictive value was 70.3%, the negative predictive value was 69%, the positive likelihood ratio was 2.09, and the negative likelihood ratio was 0.4.

Conclusions

The study’s findings suggest that FOXO4 protein could potentially serve as a valuable biomarker in the diagnosis of stroke in acute ischemic stroke patients.

Introduction

Ischemic stroke is a heterogeneous condition with distinct subtypes, such as atherothrombotic infarct, cardioembolic stroke, lacunar infarct, infarcts of unusual etiology, and essential cerebral infarct. These subtypes have different pathophysiological mechanisms, clinical features, and outcomes (Gasull & Arboix, 2022).

Acute ischemic stroke (AIS) is a leading cause of long-term work loss in developed countries and one of the most common causes of death worldwide. It is defined as the sudden loss of blood flow to a region of the brain and the resulting loss of neurological function (Phipps & Cronin, 2020). In cases of ischemic stroke, there is an interaction between unchangeable risk factors such as age and gender and modifiable risk factors such as hypertension and high cholesterol (Liu et al., 2021). The most common risk factors for ischemic stroke include hypertension, smoking, hyperlipidemia, and type 2 diabetes (Balci et al., 2011). Genetic risk factors, which are among the unchangeable risk factors for ischemic stroke, are an essential issue. For example, angiotensin-converting enzyme gene variants are associated with hypertension, while hereditary predisposition to type 2 diabetes has also been demonstrated (Alawneh et al., 2020). Genetic predisposition to ischemic stroke has been shown in animal studies and various epidemiological studies on families and twins (Della-Morte et al., 2012).

Forkhead box protein O (FOXO) genes, which have four members: FOXO1, FOXO3, FOXO4, and FOXO6, have been identified in chromosomal translocation studies on tumors (Link, 2019). The FOXO4 protein, encoded by the FOXO4 gene, is thought to be important in many cellular pathways, including oxidative stress signaling, longevity, insulin signaling, cell cycle progression, and apoptosis. Knockdown of the FOXO4 protein triggers cell proliferation and inhibits cellular apoptosis by reducing oxidative stress after cerebral ischemia/reperfusion injury (vander Horst & Burgering, 2007). On the other hand, Ep300 is an enzyme encoded by the EP300 gene in humans. EP300 is associated with multiple transcription factors involved in critical cellular processes such as apoptosis and DNA repair and may be necessary in ischemic stroke (Barrera-Vázquez et al., 2022). Our study aimed to investigate the diagnostic value of FOXO4 and Ep300 proteins in acute ischemic stroke patients admitted to the emergency department.

Materials & Methods

Patient selection

A total of 39 stroke patients and 40 healthy individuals were included in the study. The groups were divided into two groups: patient and control. For all patients participating in our study, written and verbal voluntary consent forms were obtained from themselves or their legal guardians. Age and gender were taken into consideration in patient and control groups. When selecting patients, patients who were diagnosed with acute ischemic stroke, over the age of 18, and who gave consent to participate in the study themselves or their first-degree relatives were selected. In the control group, patients who were over the age of 18 and who had no previous ischemic disease (e.g., myocardial infarction, ischemic stroke, etc.) were selected by signing a voluntary consent form. Patients were excluded if they had hemorrhagic stroke, transient ischemic attack, recent infection, autoimmune disease, malignancy, or chronic inflammatory conditions. Control individuals had no prior history of cerebrovascular disease and were not on anti-inflammatory or anticoagulant therapy. Ischemic stroke subtypes were not classified in our dataset.

Blood and tissue sample preparation

Three ml blood samples were taken from patients diagnosed with AIS in biochemistry tubes. Blood samples were immediately transferred to the Genome and Stem Cell Center for study.

Serum isolation

Blood samples were centrifuged at 4,000 g for 20 min to separate the serums, and serum samples were stored at −80 C until the study was performed.

Enzyme-linked immunosorbent assay

The Enzyme-linked immunosorbent assay (ELISA) method was used to determine the amount of Ep300 and FOXO4 protein in the sera obtained from all participants. The standards in the ELISA kit were prepared by diluting as requested by the manufacturer. The procedures were performed: 50 µL of standard solution wells, 40 µL of serum sample, and 10 µL of studied protein antibodies were added to the sample wells. Then, 50 µL of streptavidin-horseradish peroxidase was added to each well except the blank well, and the plate was covered with a leak-proof membrane. The plate was gently shaken to mix the content and incubated at 37 °C for 60 min away from light. The washing solution in the kit was diluted 25-fold. The plate was carefully washed five times. Following this, 50 µL of chromogen reagent A was added to each well, and then 50 µL of chromogen reagent B was added to each well, and the plate was incubated at 37 °C for 10 min. Finally, 50 µL of stop solution was added to each well, and the mixture’s color was changed from blue to yellow. Each well’s optical density (OD) was measured under a wavelength of 450 nm within 10 min after adding the stop solution. The linear regression equation of the standard curve was calculated according to the concentrations of the standards, the corresponding OD values, and the determined protein concentrations of the samples. All serum samples were analyzed in duplicate. Protein levels were measured in (ng/ml).

Statistical method

Data were analyzed statistically with the IBM SPSS 27 (IBM Corp., Armonk, NY, USA) program. The Kolmogorov–Smirnov test was used to determine that the distributions of variables were normal. The descriptive statistics of continuous variables gave mean values, standard deviations, and minimum and maximum values. Qualitative variables were presented as percentages and frequency. In comparisons between the two groups, the Independent Samples T-test was used as analysis when there was a normal distribution, and the Mann–Whitney U-test was used when there was no normal distribution. Differences between categorical variables were analyzed with the Chi-square test. Receiver operator characteristic (ROC) analyses were performed for FOXO4 and Ep300 proteins. The area under the ROC curve (AUC), sensitivity, specificity, positive predictive value (PPV), negative predictive value (NPV), positive likelihood ratio (LR+), negative likelihood ratio (LR-), and 95% confidence interval (CI) values were given as descriptive statistics. Statistical significance was accepted as p < 0.05.

Ethics approval

This study was conducted with local ethical approval from Erciyes University Clinical Research Ethics Board (Approval no: 2023/192, Approval date: 29.03.2023).

Results

The study was conducted on 39 acute ischemic stroke patients, 17 females and 22 males, with a mean age of 66.9 ± 11 years; the control group included 40 healthy individuals, 17 females, and 23 males, and the control group was 66 ± 11.6 years. The clinical and demographic characteristics are shown in Table 1.

Table 1 The characteristics of the patient group.

Variable	Statistics
(n = 39)
n (%)	
Female	17 (43.6)	
Age, years, mean ± SD	66.9 ± 11	
Comorbidity		
Type 2 diabetes	15 (28.5)	
Hypertension	14 (35.9)	
Cerebrovascular disease	6 (15.4)	
CAD	6 (15.4)	
CHF	1 (2.6)	
CKD	3 (7.7)	
ECG		
AF	4 (10.3)	
NSR	35 (89.7)	
Notes.

Values are expressed as n (%), mean ± SD.

CAD Coronary Artery Disease

CHF Congestive Heart Failure

CKD Chronic Kidney Disease

ECG Electrocardiogram

AF Atrial fibrillation

NSR Normal sinus rhythm

Pulse, systolic and diastolic blood pressure, respiratory rate, white blood cell, and FOXO4 protein levels were found to be significantly higher in stroke patients compared to the control group (p = 0.038, <0.001, <0.001, <0.001, <0.001 and 0.004, respectively). Hemoglobin, body temperature, and ALT values were found to be significantly lower in stroke patients compared to the control group (p = 0.004, <0.001, and <0.001, respectively) (Table 2).

Table 2 Comparison of several variables in patients and controls.

Variable	Group	Total
Mean ± SD	p	
	Control (n = 40)
Mean ± SD	Patient (n = 39)
Mean ± SD			
Age, years	66 ± 11.6	66.9 ± 11	66.5 ± 11.3	0.690	
Female, n (%)	17 (42.5)	17 (43.6)	34 (43)	0.922	
Pulse beats/ minute	81.8 ± 8.5	86.1 ± 14.9	83.9 ± 12.2	0.038	
Hemoglobin (g/dl)	13.2 ± 1.4	12.7 ± 2.5	12.9 ± 2.0	0.004	
Sodium, mmol/L	138.1 ± 2.9	136.2 ± 3.4	137.2 ± 3.3	0.764	
Chlor, mmol/L	104.8 ± 12.0	130.2 ± 7.5	104.0 ± 9.9	0.689	
Calcium, mmol/L	9.4 ± 0.6	8.9 ± 0.8	9.2 ± 0.7	0.152	
	Control (n = 40)
Median (IQR)	Patient (n = 39)
Median (IQR)	Total
Median (IQR)	p	
Fever, °C	36.5 (36.4–36.7)	36.1 (36.0–36.5)	36.4 (36.0–36.6)	<0.001	
Systolic BP, mm Hg	120.5 (115.3–125.0)	130.0 (123.0–136.0)	124.0 (117.0–130.0)	<0.001	
Diastolic BP, mm Hg	69.0 (64.0–74.8)	77.0 (70.0–80.0)	74.0 (67.0–78.0)	<0.001	
Respiration rate	16.0 (15.0–17.8)	18.0 (17.0–22.0)	17.0 (15.0–19.0)	<0.001	
Glucose, mg/dl	128.0 (109.0–144.3)	125.0 (104.0–195.0)	128.0 (106.0–152.0)	0.521	
WBC, ×10 3 /μL	8.5 (7.4–9.5)	12.1 (8.5–17.9)	9.1 (7.7–12.7)	<0.001	
Platelet, ×10 3 /μL	306.0 (254.5–324.5)	267.0 (225.0–344.0)	298.0 (236.0–326.0)	0.450	
AST, U/L	26.6 (24.6–29.9)	23.2 (18.9–35.0)	26.1 (22.1–31.5)	0.052	
ALT, U/L	26.5 (23.4–34.1)	15.0 (11.0–23.0)	23.4 (15.0–32.5)	<0.001	
BUN, mg / dL	17.0 (13.4–20.4)	18.8 (15.4–22.4)	18.4 (14.3–21.5)	0.384	
Creatine, mg/dL	0.84 (0.65–1.17)	0.89 (0.80–1.13)	0.86 (0.74–1.13)	0.152	
Potassium, mmol/L	4.2 (3.9–4.6)	4.3 (3.9–4.7)	4.2 (3.9–4.6)	0.600	
Ep300, ng/ml	366.5 (281.1–423.8)	390.9 (332.9–455.9)	374.4 (327.8–432.0)	0.379	
FOXO4, ng/ml	1.11 (0.81–1.39)	1.30 (1.12–1.68)	1.21 (0.99–1.49)	0.004	

The discrimination ability of FOXO4 and Ep300 proteins in determining the presence of stroke was measured in ROC analysis. The discriminative ability of the Ep300 protein was not statistically significant (p = 0.380). FOXO4 protein had moderate discrimination ability (AUC value = 0.705 and p = 0.002). When the cut-off value for FOXO4 protein was accepted as >1.15, its sensitivity was 74.29%, specificity was 64.52%, positive predictive value was 70.3%, negative predictive value was 69%, positive likelihood ratio was 2.09 and negative likelihood ratio was 0.4 (Figs. 1 and 2) (Table 3).

Figure 1 ROC curve of Ep300 (ng/mL).

Figure 2 ROC curve of FOXO4 (ng/mL).

Table 3 ROC analysis for diagnosis of stroke for Ep300 and FOXO4.

		AUC
(p-value)	Cut-off	Sensitivity	(% 95 CI)	Specificity	(% 95 CI)	LR+	LR-	PPV	NPV	
Stroke	Ep300	0.563
(0.380)	>329.37	82.86	66.4–93.4	35.48	19.2–54.6	1.28	0.48	59.2	64.7	
FOXO4	0.705
(0.002)	>1.15	74.29	56.7–87.5	64.52	45.4–80.8	2.09	0.40	70.3	69.0	

Discussion

The most prominent finding of this study was that FOXO4 protein levels were significantly elevated in AIS patients compared to healthy individuals. While our preliminary study demonstrates a possible association between FOXO4 and stroke, we believe it is too early to conclude that FOXO4 has diagnostic value for stroke.

Stroke continues to be the third leading cause of death and disability worldwide. The limited availability of biomarkers in the diagnosis of stroke remains a significant challenge. Since no biomarker has demonstrated adequate sensitivity, specificity, rapidity, accuracy, and cost-effectiveness, attempts to discover new markers are paramount (Kamtchum-Tatuene & Jickling, 2019). Various markers associated with stroke pathogenesis, such as cellular energy, inflammation, and cell death regulated by microRNAs, may be reliable markers for risk prediction, diagnosis, and prognosis of ischemic stroke (Kadir, Alwjwaj & Bayraktutan, 2022; Rani et al., 2023).

It has been reported that the EP300 protein, an enzyme encoded by the EP300 gene, plays a role in regulatory T-cell function and is expected to be used in possible cancer immunotherapy. No relationship has been shown between EP300, which is thought to be effective in the molecular mechanism of ischemic stroke, and stroke (Wang et al., 2022). Similarly, there was no significant difference between the stroke and control groups regarding EP300 protein levels in our study. However, the sample size in our study was small, and our results cannot be generalized. Hence, these findings should be interpreted cautiously.

To date, >55 FOX proteins have been identified in mammals, including humans, and are divided into 19 subfamilies (FOXA to FOXS). All FOX family members are involved in various cellular functions, including embryological development, maintenance of cellular homeostasis, metabolism, and aging. The FOXO subfamily is interesting due to its versatile roles in different biological functions, including cellular metabolism, cellular proliferation, stress tolerance, cellular survival, and tumor suppression (Rani et al., 2023). The FOXO family includes four members expressed in almost all tissues: FOXO1, FOXO3, FOXO4, and FOXO6. It is known that FOXO4 protein expression increases in the brain after traumatic brain injury. Although the role and mechanism of FOXO4 in cerebral ischemia are not established, an in vitro cellular injury model study has shown that suppression of FOXO4 reduces cerebral microvascular endothelial cell damage (Cui, Li & Yuan, 2024). In our research, FOXO4 protein levels were statistically significantly higher in patients with acute ischemic stroke compared to the control group. In addition, the FOXO4 protein had a moderate discriminatory ability in predicting stroke, and its sensitivity was over 70%.

The main limitation of our study is the small study population and the fact that it was conducted in a single center. Secondly, the case-control design, where disease status was known at the time of enrollment, may introduce selection bias. Additionally, another limitation was that we used healthy controls, but in real-life emergency settings, there were stroke mimics (e.g., seizures, intracerebral hemorrhage). In future studies, control groups should include differential diagnoses, such as stroke mimics. Lastly, we did not classify ischemic stroke subtypes in our study, and the value of our results for all types of ischemic stroke is uncertain.

Conclusions

Our study investigated the diagnostic value of FOXO4 and Ep300 proteins in patients admitted to the emergency department with acute ischemic stroke. FOXO4 protein had a significantly high and moderate discriminatory ability in stroke patients. Considering the small sample size and some limitations of our study, FOXO4 may have a promising but mild diagnostic value for stroke. We believe that future comprehensive studies with larger patient numbers are needed to investigate the value of FOXO4 in ischemic stroke patients.

Supplemental Information

Supplemental Information 1 Clinical characteristics and laboratory findings of the control and patient groups

Supplemental Information 2 STROBE Statement

Additional Information and Declarations

Competing Interests

Author Contributions

Human Ethics

Data Availability

The authors declare there are no competing interests.

Emre Bülbül conceived and designed the experiments, performed the experiments, prepared figures and/or tables, and approved the final draft.

İbrahim Toker performed the experiments, analyzed the data, prepared figures and/or tables, and approved the final draft.

Halime Dana conceived and designed the experiments, authored or reviewed drafts of the article, data curation, and approved the final draft.

Mehmet Doğan performed the experiments, authored or reviewed drafts of the article, and approved the final draft.

İbrahim Mercan analyzed the data, authored or reviewed drafts of the article, and approved the final draft.

Burcu Çoşkun Işık analyzed the data, prepared figures and/or tables, and approved the final draft.

Elif Funda Şener conceived and designed the experiments, authored or reviewed drafts of the article, and approved the final draft.

The following information was supplied relating to ethical approvals (i.e., approving body and any reference numbers):

Erciyes University Clinical Research Ethics Board (2023/192) approved this research.

The following information was supplied regarding data availability:

The raw data is available in the Supplemental File.

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
