# Peer review of "Diagnostic value of Ep300 and FOXO4 proteins in acute ischemic stroke: preliminary findings from a case-control study"

_PeerJ, doi:10.7717/peerj.19970_

## Round 0.1 · original submission · Major Revisions

Please address the concerns of both reviewers and amend the manuscript accordingly.

·

Basic reporting

The authors present the results of a single-center case-control clinical study aimed at analyzing the diagnostic value of FOXO4 (Forkhead box protein O) and Ep300 proteins in patients with acute ischemic stroke. The study included a total of 39 acute ischemic stroke patients and 40 healthy individuals (control group). The authors suggest that Foxo4 protein could potentially serve as a valuable biomarker in the diagnosis of stroke in acute ischemic stroke patients. The paper is potentially interesting, but some aspects of the manuscript may be improved by taking into account the following points.

Experimental design

Original primary research within the aims and Scope of the journal.
Methods are described with sufficient detail to replicate.

Validity of the findings

Conclusions are well stated, linked to the original research question.

Additional comments

1) Please homogenize the terms used in the Abstract and the text (ex, “Foxo4”, “FOXO4”, etc.).

2) Due to the small size of the study, the title should clearly mention “preliminary findings.”

3) In the Introduction section, it should be emphasized that ischemic stroke is a heterogeneous disease and that, in clinical studies, it is extremely interesting to adequately differentiate the different stroke subtypes (atherothrombotic infarct, cardioembolic stroke, lacunar infarct, infarct of unusual etiology, essential cerebral infarct) among the population with stroke. This recommendation is due to the impact of stroke subtypes on the distribution of risk factors, stroke severity, and outcome (Int. J. Mol. Sci. 2022, 23(16), 9476; https://doi.org/10.3390/ijms23169476).

4) It would be interesting to know the different acute ischemic stroke subtypes in the study population.

5) We recommend that the authors begin the Discussion by evaluating the most relevant data from their study.

6) It would be interesting to note, as a possible future line of research, if these results were also confirmed in the different ischemic stroke subtypes, mainly in the subgroup of patients with lacunar infarcts. In this subgroup, the main cardiovascular risk factors are hypertension and diabetes, and the pathophysiology, prognosis, and clinical features of acute small vessel disease strokes are different from all other cerebral infarcts (see and add this reference; BMC Neurol 2010; May 18;10:31. doi: 10.1186/1471-2377-10-31).

7) The opinion of the authors on future lines of research on this topic should be added in the text.

8) Please check reference #9

Reviewer 2 ·

Basic reporting

-

Experimental design

The study aims to investigate the diagnostic value of FOXO4 and Ep300 proteins in acute ischemic stroke (AIS) patients presenting to the emergency department, which falls within the scope of a diagnostic study. While this is a clinically relevant topic, several important methodological limitations need to be addressed to strengthen the validity of the findings:

1. Study design limitations: Based on the methods section, the study included 39 stroke patients and 40 healthy individuals, with the disease status already known at the time of enrollment. For diagnostic accuracy studies, it is essential that participants are enrolled before the final diagnosis is known, to avoid selection bias. This is a major limitation of the current design.

2. Inclusion/exclusion criteria: The inclusion and exclusion criteria for both the stroke and control groups are not clearly defined, which affects the reproducibility and interpretation of the findings.

3. Choice of control group: The control group comprises healthy individuals, whereas in real-world emergency settings, the clinical challenge lies in distinguishing AIS from other acute neurological conditions, such as intracerebral hemorrhage, seizures, or hypoglycemia. We strongly recommend replacing the healthy control group with an age- and sex-matched cohort of emergency patients with common stroke mimics to enhance the clinical applicability of the study.

4. Sample size: The overall sample size is relatively small, which limits the statistical power and generalizability of the results.

Validity of the findings

Interpretation of diagnostic performance: According to the reported AUC values, the claims of "significantly high and good discriminatory ability" are not well-supported. A more cautious interpretation is warranted.

---

## Round 0.2 · Minor Revisions

Please address the remaining concerns of Reviewer #2 and amend the manuscript accordingly.

·

Basic reporting

I appreciate the authors efforts to include rhe reviewer’s comments. I do not have further comments.

Experimental design

The research question is well defined and the investigation performed is rigorous.

Validity of the findings

The responses to the reviewer’s comments are clear and well-justified, and the additional data provided further support the conclusions.

Additional comments

The manuscript has been properly revised. It is acceptable

Reviewer 2 ·

Basic reporting

no comment

Experimental design

no comment

Validity of the findings

The authors have revised the manuscript based on the previous review comments. However, I would like to make the following two additional suggestions:

Given the limitations in study design and sample size, the conclusions should be drawn with extreme caution. The findings of this study can only suggest that FOXO4 may have some diagnostic value for stroke.

Additionally, as stated in the first sentence of the Discussion section, this study merely indicates that FOXO4 protein levels are higher in stroke patients compared to controls, suggesting a potential association between FOXO4 and stroke. It is premature to conclude that FOXO4 has diagnostic value for stroke.

---

## Round 0.3 · accepted · Accept

Since the remaining issues pointed by the reviewer were adequately addressed, the revised manuscript is acceptable now.